# Propagule Dispersal Determines Mangrove Zonation at Intertidal and Estuarine Scales

**Wenqing Wang, Xiaofei Li and Mao Wang ***

Key Laboratory of the Coastal and Wetland Ecosystems, Ministry of Education, College of the Environment & Ecology, Xiamen University, Xiamen 361102, China; mangroves@xmu.edu.cn (W.W.); lixiaofei819@163.com (X.L.)
* Correspondence: wangmao@xmu.edu.cn; Tel.: +86-136-6603-7893

**Abstract:** Propagule dispersal has generally been recognized as a vital factor affecting the spatial structure of tropical forest plants. However, available research shows that this hypothesis does not apply to mangrove species the propagules of which are dispersed by water. Due to the lack of comprehensive and quantitative information as well as the high spatio-temporal heterogeneity of the mangrove environment, the exact factors affecting the spatial structure of mangrove forests are poorly understood. To assess this, we selected a mangrove estuary with high mangrove species richness that experiences great changes in water salinity. After investigating the zonation of mature mangrove individuals across tides and the estuary, we measured the size and initial specific gravity of the propagules and then selected the eight most common species from which to observe the changes in specific gravity, buoyancy, and root initiation during dispersal at different sites with different water salinity regimes. The relationships among distribution patterns, propagule establishment, and dispersal behavior were investigated. We found that mangrove propagule dispersal is not a passively buoyant process controlled by water currents. During dispersal, mangrove propagules can actively adjust their specific gravity and root initiation. The dynamic specific gravity of the propagules was negatively related to propagule buoyancy and surface elevation. The differences in propagule specific gravity corroborated the distribution patterns of the species across the intertidal zone and estuary. Mangrove zonation on both the intertidal and estuarine scale can be explained by the tidal sorting hypothesis, as zonation is controlled by the tidal sorting of the propagules according to buoyancy and by the differential ability of the propagules to establish in the intertidal zones. The results add new understanding of observed mangrove species zonation and should inform conservation managers when restoring mangroves or evaluating the potential impacts of global change and anthropogenic disturbances that might alter the hydrology, including the water salinity regime.

**Keywords:** buoyancy; dispersal; specific gravity; establishment; mangrove; propagule; water salinity; zonation

## 1. Introduction

Propagule dispersal, as the movement of organisms leading to gene flow, has generally been recognized as a vital factor affecting population dynamics and community structure [1–4]. This has been proven for marine fish, benthos [5–9], and many terrestrial plant communities, especially tropical forests [10–12]. Additionally, propagule dispersal has been increasingly recognized as an essential process in shaping the effects of climate change on the distribution of species [13,14].

As for higher plants, the principal abiotic agents of propagule dispersal are wind and water [15]. However, available research on propagule dispersal has focused on species passively dispersed by wind, and theoretical models of seed dispersal have been well established [4,16,17]. In contrast, research on propagule dispersal by water is uncommon [17,18].

Mangrove forests are well known for their spatial distribution (zonation) of species along environmental gradients [1,19–21]. Locally, mangrove zonation can be conveniently considered at two orthogonal scales or in two ways: estuarine (from the river mouth to the upstream penetration distance of saltwater) and intertidal (from the low intertidal zone to the high intertidal zone) [21–23]. Specific mangrove species will occupy one or all of the three parts according to their salt tolerance and other biotic or abiotic factors. For example, *Sonneratia alba* J. Smith. occupies downstream areas and is thus classified as a downstream group [19,24], while *Sonneratia caseolaris* (L.) Engl. can grow upstream at the penetration distance of salt water and is thus classified as an upstream group [25]. On an intertidal scale, one of the most striking features of mangrove forests is the characteristic zonation whereby mangrove species tend to distribute differentially in a banded zonation pattern orientated roughly parallel to seashores from approximately mean sea level to the highest spring tides [22,26].

Understanding the determinants of zonation is key to elucidating the structure and function of mangroves under sea-level rise and anthropogenic disturbance [24]. The zonation pattern of mangrove species has been a research focus for decades, especially on an intertidal scale, and has led to numerous hypotheses attempting to explain such zonation. However, much controversy still exists regarding this topic [1,24,27–31].

Most mangrove species have buoyant, water-borne propagules, and some propagules can drift by ocean currents for a few months and for thousands of kilometers [14,21]. In some studies, the specific gravity of the mangrove propagules during dispersal has been arbitrarily considered unchangeable [14,18,30,32], and plastic drift cards or artificial drogues have been used as indicators of propagule dispersal [32]. However, in contrast to most terrestrial plants, most mangrove propagules do not hibernate [21]. During dispersal, mangrove propagules experience a series of physio-chemical changes. After transplantation into solutions with different salinities, *Kandelia obovata* Sheue, Liu et Yong propagules exhibited slow ion exchange with the surrounding water, which was controlled by water salinity [33]. A few studies have reported the influence of water salinity on the dispersal ability of mangrove propagules [23,31,34,35]. *Aegiceras corniculatum* (L.) Blanco propagules can remain buoyant in full seawater for up to three months but will sink in brackish water within one week [36].

Mangrove zonation is typically indicated by co-living species with different types of propagules, and thereafter by species with different dispersal and establishment strategies [34]. The shape and size of mangrove propagules vary widely across species [21,23,31]. Despite belonging to the same family (Rhizophoraceae) and sharing the same reproductive strategy (vivipary), the two co-living species, *Ceriops tagal* (Perr.) C.B. Rob. and *Rhizophora mucronata* Lam., have different dispersal and establishment strategies [34]. This condition also occurs in *Bruguiera gymnorhiza* (L.) Lam. and *Rhizophora stylosa* Griff. [37]. Therefore, it is not appropriate to extrapolate a general conclusion from a few species-specific studies [38,39]. However, almost all of the available reports draw conclusions on the basis of the investigation of a few species [1,24,30,34,40]. Clarke et al. [23] conducted a classic experiment aiming to determine the relationship between early life history traits of mangroves and adult distribution patterns. Unfortunately, the report did not include species of *Sonneratia*, which has a small seed size and low-intertidal distribution [19]. Previous studies have failed to include a variety of mangrove propagules, which explains why a widely applicable explanation has not yet been found [39].

De Ryck et al. [30] suggested that other environmental characteristics, such as palatability, buoyancy, number of released propagules and tidal position, rather than propagule size, influence dispersal capacity. Tidal flooding, land elevation, and salinity are often considered as controlling factors in mangrove zonation [37]. Jiménez and Sauter [38] suggested that the causal factors determining species zonation change with species and sites, and that no general explanation exists. It has also been proven that predation by crabs on floating propagules cannot account for the zonation of mangrove species [28].

On an intertidal scale, surface elevation and related variables, such as hydroperiod, soil salinity, and soil physical-chemical characteristics, have been recognized as major drivers of zonation [19,21,24].

Mangrove zonation is complex due to the interactional effect of biotic and abiotic factors on the distribution and survive of mangrove plants. Additionally, due to the lack of comprehensive and quantitative information regarding the distribution patterns of mangrove species along intertidal zones or estuaries, mangrove zonation on such scales has rarely been examined [23].

We hypothesized that the dispersal of mangrove propagules is not a passive dispersal process, and that mangrove zonation is determined by the buoyancy of the propagules. We also hypothesized that the propagule can actively adjust its gravity and root initiation during dispersal according to its environment, especially water salinity.

## 2. Materials and Methods

### 2.1. Research Site

The study was conducted at Bamen Bay (19°22′–19°35′ N, 110°40′–110°48′ E), a mangrove estuary situated in the northeastern part of Hainan Island, China (Figure 1). The bay has a tropical monsoon climate, with a mean annual rainfall of 1974 mm and temperature 24.0 °C. It is a semi-enclosed estuary with a mangrove area of 2000 ha, and is fed by two small rivers. The bay is subjected to irregular diurnal tides with a mean tidal amplitude of 0.75 m. A total of 25 mangrove species and several associates occur naturally in the bay, and the bay is recognized as the bay with the highest mangrove species richness in China [25]. The Wenchang River is a small river feeding into the bay, and riverine mangroves grow well along the river. Therefore, the natural occurrence of all types of propagules in the same estuary provides us with a rare opportunity to research the factors determining the estuarine and intertidal distribution patterns of mangrove species.

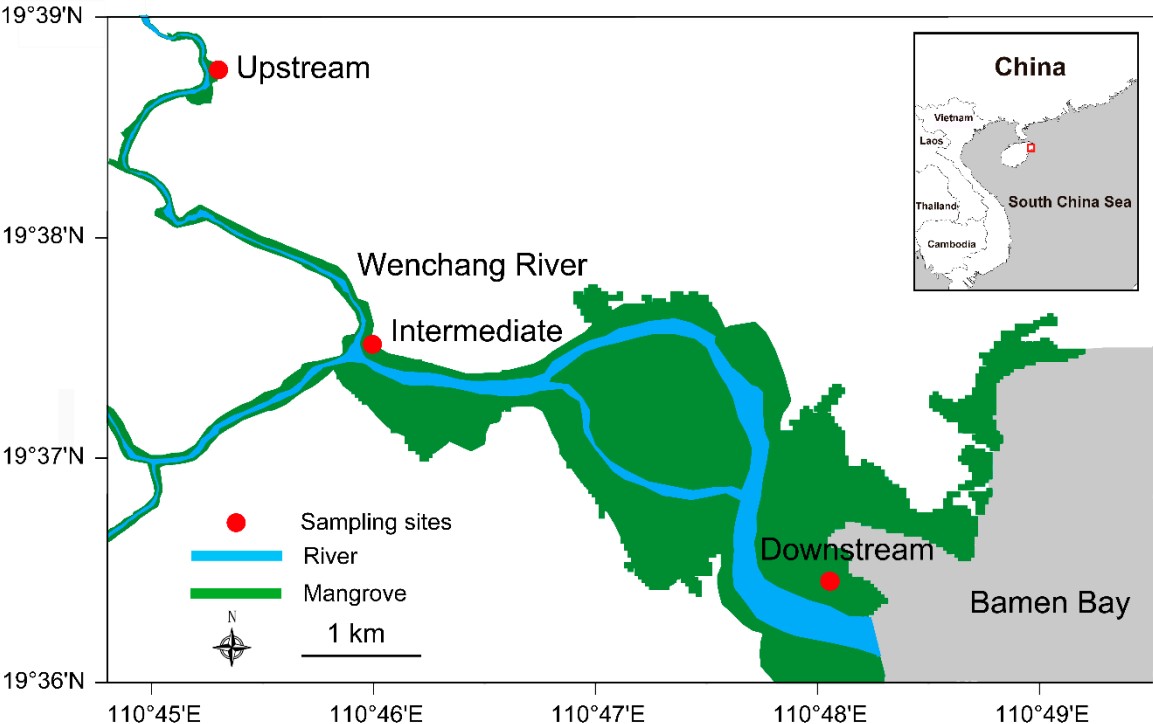

**Figure 1.** Map of the study area (Bamen Bay, Hainan, China) showing the mangrove distribution and the locations of sites surveyed.

From the river mouth to the upstream limit of the mangrove distribution, we divided the bank of the Wenchang River into three sections: upstream, intermediate, and downstream according to Duke et al. [22]. Three sites with different salinity regimes were selected to measure the changes in propagule characteristics during dispersal and buoyancy. The upstream site was located in the

upstream section of the river. This mangrove community was mainly composed of *A. corniculatum*, *S. caseolaris* and *Bruguiera sexangula* (Lour.) Poir. The water salinity in the rainy season was close to zero (0.1‰–0.2‰). The intermediate site was located at the intermediate section of the river with a water salinity of 15‰–25‰. The mangroves were dominated by *A. corniculatum*, *B. sexangula* and *B. gymnorhiza*. The downstream site was located at the downstream section of the river. The mangroves at the downstream site were the least influenced by fresh water and were mainly composed of higher salt-tolerant species such as *S. alba*, *B. sexangula*, *Lumnitzera racemosa* Willd., *R. stylosa*, and *C. tagal*. Figure 2 shows the daily water salinity regimes of the three sites in the rainy season.

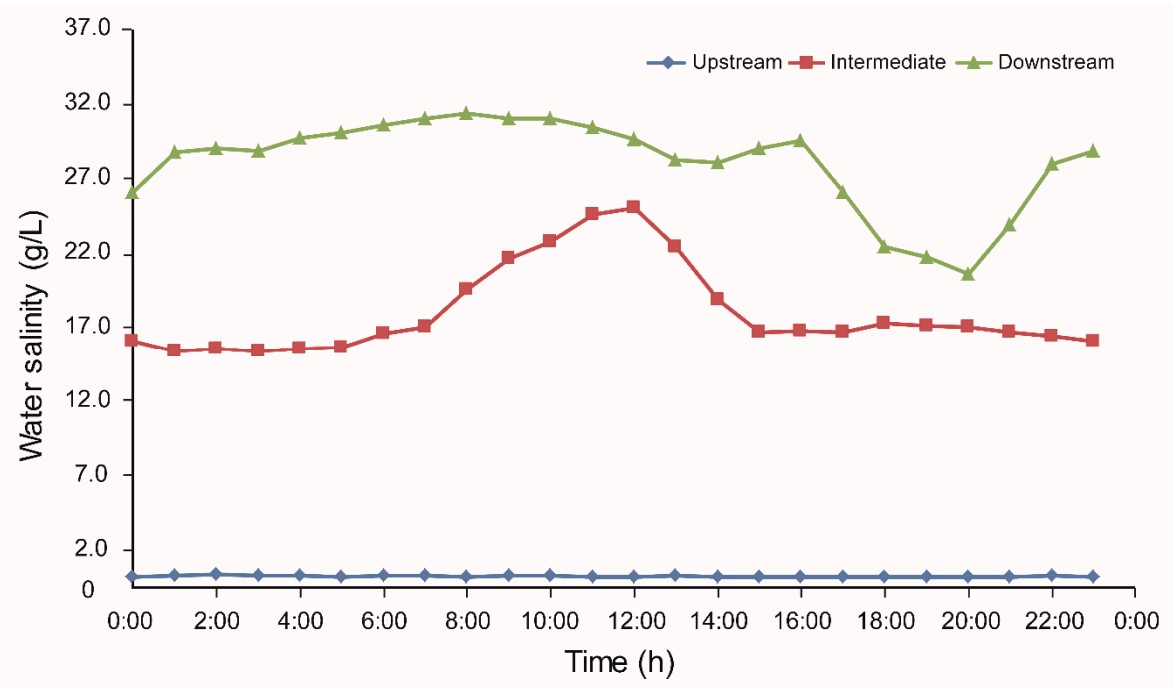

**Figure 2.** Typical diurnal change of the surface water salinity at the upstream site, intermediate site and downstream site of the Wenchang River, Hainan, China.

### 2.2. Measurement of Surface Elevation

Six transects perpendicular to the shore of the Wenchang River were established and numbered sequentially. Transect I and Transect II were located at the upstream section in the vicinity of the upstream site. Transect III and Transect IV were located at the intermediate section and were close to the intermediate site. Transect V and Transect VI were located at the downstream section and close to the downstream site (Figure 1). A series of survey stations (15 to 30 m apart) were arranged seaward to landward along each transect. The total length of each transect ranged between 70 m and 300 m. Each station was numbered consecutively seaward to landward. In this research, the landward area of the mangroves was defined as the intersection of mangroves and terrestrial vegetation, and the plant community was dominated by associates, including *Heritiera littoralis* Aiton, *Hibiscus tiliaceous* L., and *Barringtonia racemosa* (L.) Spreng.

At each survey station, three 10 m × 10 m quadrats parallel to the shore were established. The distance between adjacent quadrats was 20 m. The number of mature individuals of each mangrove species in each quadrat was recorded. Trees taller than 1.8 m and shrubs taller than 0.6 m were assumed to be mature individuals [41]. The relative elevation of each quadrat was measured using a GPS-RTK device (real-time kinematic, Trimble Navigation Limited, USA, vertical and horizontal accuracies ± 8–15 mm). Considering differences in the surface elevations of the seaward mangrove forests along each transect, the surface elevation of the seaward forest of Transect I was assumed to be 0 cm. This allowed for comparisons between transects.

The surface elevation (SE) of a mangrove species in a transect was calculated as follows:

$$\text{SE} = \frac{\sum_i^K (N_i \times H_i)}{\sum N_i} \tag{1}$$

$i$: code number of each survey station

$K$: code number of the landward station of a transect

$H_i$: surface elevation of station i

$N_i$: individual number of mangrove species at station i

This calculation allows any species occurring in a transect to be assigned an elevation value [42]. This method assumes that the same mangrove species in any quadrat has the same surface elevation. It provides a high resolution, straightforward and cost-effective method to measure the surface elevation of mangrove species at high spatial resolution. This method is similar to the method used by Leong et al. [43] and Oh et al. [42].

*2.3. Collection of Mature Propagules*

The mature propagules of eight mangrove species recorded in the six transects were collected in the bay from April to November, 2008 (Table 1). The dispersal units of mangroves are classified into four groups: spore, seed, fruit and seedling [21]. We considered a dispersal unit as a propagule in this study. The propagules of the mangrove species of *Sonneratia* are generally regarded as multiple-seeded fruits [31,44]. However, according to our field observation, the green fruits that fall from parent trees may initially be buoyant for 1–2 days and can be carried some distance by the tides. Following this, the pulp rots and numerous small angular seeds are released. Therefore, the seed was regarded as the propagule for *S. alba* and *S. caseolaris* in this study. The propagules of *A. corniculatum* and *L. racemosa* are one-seeded fruits. Therefore, an individual fruit was regarded as the propagule. *B. gymnorhiza*, *B. sexangula*, *C. tagal*, and *R. stylosa* are true viviparous mangroves [21]. Their embryos grow and break through the seed coats and then exit the fruit walls while still being attached to the parent plants. They finally transform into a rod-like shape with a pointed end. The propagules are not seeds but seedlings. For these four species, the individual seedling (hypocotyl) was regarded as the propagule. All of the mature propagules were collected from adult trees during peak periods of propagule release for each species by gently shaking the branches and collecting the released propagules to prevent exposure to seawater once they had fallen. The selected propagules were of uniform age, and those that were damaged by boring insects or crabs were discarded. The propagules of *A. corniculatum* were collected with the pericarp.

**Table 1.** Type, size, initial specific gravity, and final dynamic specific gravity of the propagules (*n* = 20) of eight mangrove species after 20 days of floating along the Wenchang River, Hainan, China. Nomenclature follows Tomlinson [21]. Values are mean ± SE. Different letters after specific gravity data of the same species indicate significant differences at *p* < 0.05.

| Species | Propagule Type | Size (Fresh Weight) (g) | Initial Specific Gravity (g/cm$^3$) | Final Dynamic Specific Gravity (g/cm$^3$) | | |
|---|---|---|---|---|---|---|
| | | | | Upstream | Intermediate | Downstream |
| *Sonneratia caseolaris* (L.) Engl. * | Seed | 0.006 ± 0.017 | 0.810 ± 0.060 a | 0.985 ± 0.024 b | 1.031 ± 0.038 c | 1.080 ± 0.048 c |
| *Lumnitzera racemosa* Willd. | One-seeded fruit | 0.186 ± 0.028 | 0.868 ± 0.043 a | 0.963 ± 0.014 b | 0.916 ± 0.021 c | 0.948 ± 0.010 d |
| *Sonneratia alba* J. Smith. * | Seed | 0.072 ± 0.017 | 0.913 ± 0.043 a | 1.113 ± 0.031 b | 1.116 ± 0.020 b | 1.124 ± 0.025 c |
| *Aegiceras corniculatum* (L.) Blanco | Fruit/Cryptic hypocotyl | 0.907 ± 0.130 | 0.966 ± 0.026 a | 1.077 ± 0.009 b | 1.070 ± 0.010 b | 1.105 ± 0.019 c |
| *Ceriops tagal* (Perr.) C.B. Rob. | Seedling/Hypocotyl | 7.378 ± 1.277 | 0.975 ± 0.010 a | 0.978 ± 0.007 a | 0.991 ± 0.009 b | 1.004 ± 0.002 b |
| *Bruguiera sexangula* (Lour.) Poir. | Seedling/Hypocotyl | 9.252 ± 1.133 | 0.978 ± 0.027 a | 1.000 ± 0.005 b | 1.005 ± 0.005 b | 1.008 ± 0.001 c |
| *Bruguiera gymnorhiza* (L.) Lam. | Seedling/Hypocotyl | 23.742 ± 2.101 | 0.994 ± 0.011 a | 1.014 ± 0.008 b | 1.008 ± 0.002 b | 1.025 ± 0.004 c |
| *Rhizophora stylosa* Griff. | Seedling/Hypocotyl | 16.984 ± 3.075 | 1.015 ± 0.063 a | 0.992 ± 0.001 a | 0.987 ± 0.002 b | 1.009 ± 0.007 a |
| Average | | | 0.891 | 1.015 | 1.014 | 1.038 |

* The floating time for *Sonneratia alba* and *Sonneratia caseolaris* was 15 days.

### 2.4. Observation of Specific Gravity, Buoyancy and Root Initiation during Propagule Dispersal

At each site, three floating plastic boxes (width 1 m, length 1 m, height 0.4 m, mesh 1 mm) supported by a wooded frame were arranged. To prevent disturbance from crabs, birds, or other animals, all wooded frames were covered with a polythene net (mesh 2 mm). All of the boxes were kept afloat on the water with plastic foam; the upper 20 cm was above the water surface, and the boxes were fastened to fixed stakes to prevent drifting by tides.

According to abundance, propagule type, intertidal distribution, and estuarine distribution, eight most common species were selected to observe the changes in specific gravity, buoyancy, and root initiation during dispersal. As only a few *Avicennia marina* (Forstk.) Vierh. individuals were recorded in Transect VI, this species was not selected in our floating experiment. To measure the buoyancy of the propagules, 20 propagules of each species were arranged in each floating box at the three sites mentioned above. Each site had three boxes per species, and the three boxes were divided into two groups. One group was used to observe the buoyancy and root initiation during dispersal, while the other was used to measure the dynamic specific gravity.

The buoyancy (proportion of floating propagules) was recorded after 1, 3, 5, 7, 10, 15, and 20 days. Propagule specific gravity and root initiation measurements were conducted synchronously. Three propagules were randomly collected from each box for specific gravity measurements 1, 3, 5, 7, 10, 15, and 20 days later. All of the propagules in the field trial were consistently immersed in water during the experimental periods. The observations of *S. caseolaris* and *S. alba* only lasted 15 days because most of the seeds germinated within this period.

### 2.5. Measurements of Propagule Characteristics

The initial specific gravity was defined as the specific gravity of the mature propagules measured immediately after being picked from the parent trees to prevent exposure to seawater once they had fallen. As for the species of *Sonneratia*, the initial specific gravity referred to the specific gravity of the newly released seeds. Twenty mature propagules of each species were collected during each species' propagule maturation season from April to November, 2008, for specific gravity measurements.

The buoyancy of the propagules was divided into two categories: floating and sinking. The propagules floating both horizontally on the water surface and vertically between the water surface and the bottom and that could be moved by tides, were classified as floating, whereas those sinking to the bottom were classified as sinking. Root initiation was judged by the appearance of the roots through little cracks in the root bumps or by the roots breaking through the episperm. Seeds were considered to have germinated when the radicle had protruded the seed coat by at least 2 mm.

In consideration of the irregular shapes of the propagules of the various mangroves, propagule size was expressed as the fresh weight, which was measured with an analytical balance (1 mg resolution). Propagule volume was measured by using water-displacement method according to the revised Archimedes' principle [45].

### 2.6. Statistics

The mean and standard deviation (SD) of the specific gravity and fresh weight of the propagules were calculated for each species, and their correlations were determined. Data on all propagule dispersal parameters were analyzed for differences among the three sites and among floating times by univariate analysis of variance. When the difference was significant at $p < 0.05$, a post-hoc test was used to determine the potential source of the difference. All of the analyses were performed with SPSS (Version 16.0, SPSS Inc., Chicago, IL, USA). Statistical significance was defined at $p < 0.05$.

## 3. Results

### 3.1. Static Densities of Mangrove Species

The type and initial specific gravity of the propagules of eight mangrove species are shown in Table 1. Their propagule densities ranged between 0.810 g/cm$^3$ and 1.015 g/cm$^3$ with a mean value of 0.891 g/cm$^3$, and showed significant differences ($p < 0.01$) among species.

### 3.2. Changes in Propagule Specific Gravity during Floating

During floating, with the exception of the two species of *Bruguiera*, the densities of the propagules of all the mangrove species changed dramatically with floating time ($p < 0.05$) (Figure 3 and Table 2). The densities of the propagules of *B. gymnorhiza*, *R. stylosa*, *A. corniculatum*, and *S. caseolaris* also differed significantly among sites (Figure 3 and Table 2). The specific gravity of *B. gymnorhiza* propagules increased from 0.994 g/cm$^3$ to 1.009 g/cm$^3$ at all sites (Figure 3a). At the upstream and intermediate sites, the specific gravity peaked after 7 days of floating, while at the downstream site it peaked after 15 days of floating. After 3 days of floating, the specific gravity of the propagules of *A. corniculatum* increased sharply from 0.966 g/cm$^3$ to 1.047 g/cm$^3$, and then increased progressively with floating time (Figure 3f). *S. alba* showed even quicker changes. After 1 day of floating, the specific gravity increased from 0.913 g/cm$^3$ to 1.096 g/cm$^3$ and stabilized at 1.1 g/cm$^3$ at all sites. During the floating experiments, the densities of the propagules of *C. tagal* and *L. racemosa* were both lower than 1.0 g/cm$^3$, although *C. tagal* showed an increasing trend (Figure 3b,d). According to the changes in propagule specific gravity during floating, the eight mangrove species could be divided into two categories. The first category included *B. gymnorhiza*, *R. stylosa*, *S. caseolaris* and *A. corniculatum*; the changes in their propagule densities were site specific ($p < 0.001$). However, the specific gravity of the propagules of the four other mangroves did not differ significantly among sites.

**Table 2.** Results of two-way ANOVA test ($p < 0.05$) showing the effects of site and floating time on the propagule specific gravity of eight mangrove species along the Wenchang River, Hainan, China.

| Species | Site (S) | | Floating Time (F) | | S × F | |
|---|---|---|---|---|---|---|
| | F | p | F | p | F | p |
| *Sonneratia caseolaris* | 3.899 | 0.049 * | 43.440 | 0.001 ** | 1.739 | 0.188 |
| *Lumnitzera racemosa* | 0.510 | 0.476 | 4.599 | 0.033 ** | 2.678 | 0.103 |
| *Sonneratia alba* | 0.690 | 0.407 | 36.112 | 0.001 ** | 0.289 | 0.591 |
| *Aegiceras corniculatum* | 10.696 | 0.001 ** | 57.247 | 0.001 ** | 1.396 | 0.239 |
| *Ceriops tagal* | 0.437 | 0.510 | 32.421 | 0.001 ** | 0.792 | 0.375 |
| *Bruguiera sexangula* | 0.255 | 0.614 | 1.416 | 0.236 | 1.844 | 0.176 |
| *Bruguiera gymnorhiza* | 18.351 | 0.001 ** | 0.064 | 0.801 | 3.498 | 0.063 |
| *Rhizophora stylosa* | 55.180 | 0.001 ** | 7.908 | 0.005 ** | 13.918 | 0.001 ** |

\* $p < 0.05$; \*\* $p < 0.01$.

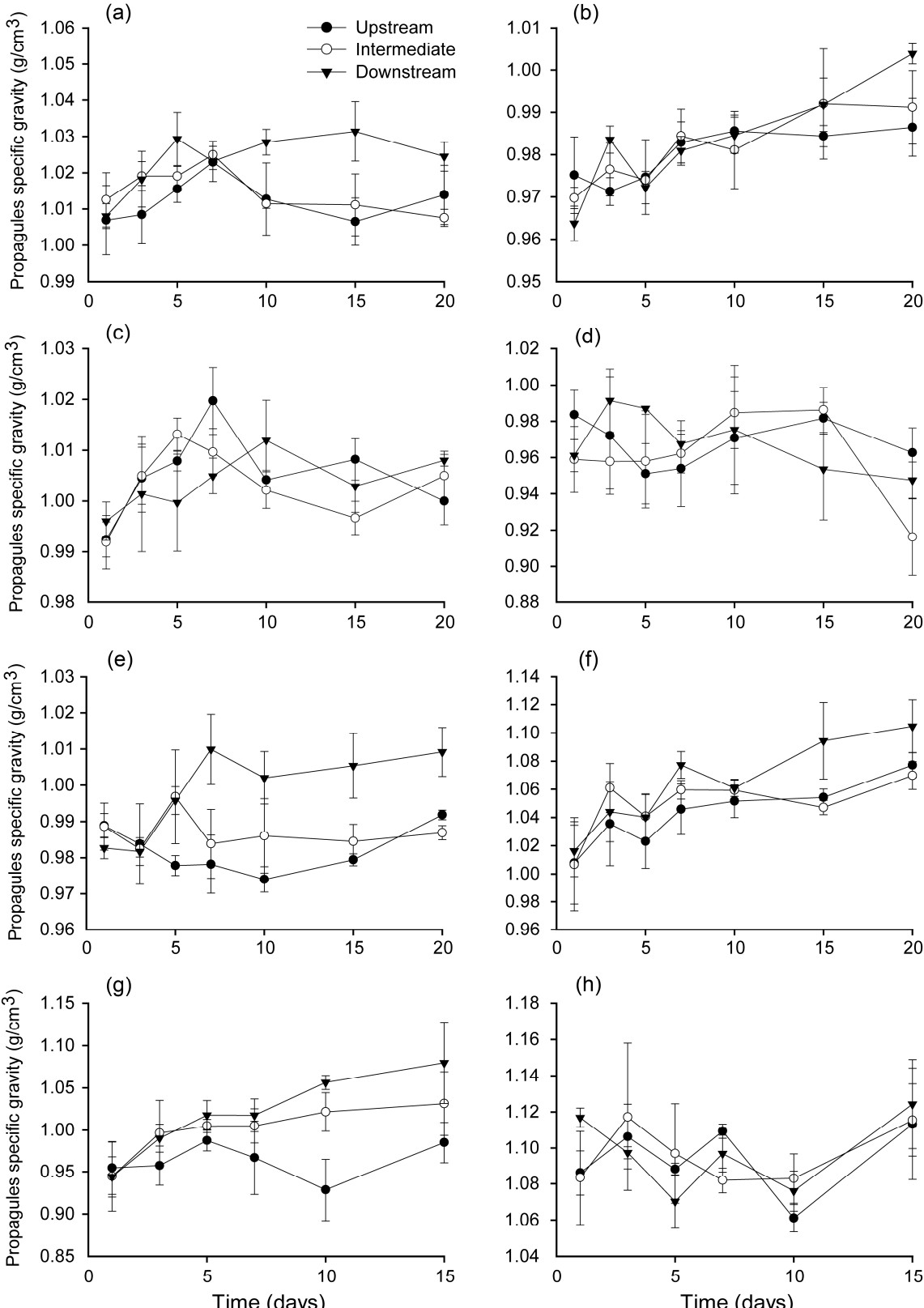

**Figure 3.** Changes in the specific gravity of the propagules of eight mangrove species during floating at three sites with different water salinity regimes. Mean specific gravity (±SE) shown from 20 propagules placed in each of three replicated grid of woody frame. (**a**) *Bruguiera gymnorhiza*; (**b**) *Ceriops tagal*; (**c**) *Bruguiera sexangula*; (**d**) *Lumnitzera racemosa*; (**e**) *Rhizophora stylosa*; (**f**) *Aegiceras corniculatum*; (**g**) *Sonneratia caseolaris*; (**h**) *Sonneratia alba*.

### 3.3. Changes in Propagule Buoyancy during Floating

Corresponding to the changes in propagule densities, the buoyancy of the propagules changed with sites and floating times, and different species showed different trends (Figure 4). The buoyancy rate of the *B. gymnorhiza* and *B. sexangula* propagules at the downstream site was significantly higher than that at both the upstream and intermediate sites ($p < 0.01$) (Figure 4a,c). More than half of the *B. sexangula* propagules had lost their buoyancy after floating for 3 days at the upstream site and intermediate site; however, approximately 90% remained afloat at the downstream site (Figure 4c). After 1 day of floating, the buoyancy rate of the *L. racemosa* propagules decreased dramatically from 100% to 75% and then remained stable (Figure 4d). The buoyancy rates of the propagules of *R. stylosa* and *C. tagal* at the intermediate site and downstream site were higher than 80% and showed few changes during the entire floating experiment (Figure 4b,e). After 1 day of floating, only 30% of the propagules of *A. corniculatum* floated at all sites, and no propagule remained afloat after 7 days (Figure 4f). During the 15 days floating experiment, only a small portion of *S. alba* propagules remained afloat (Figure 4h). In our study, all of the mangrove species except *S. caseolaris* shared a common trend, i.e., the propagules had a higher buoyancy rate at the downstream site than at the intermediate or upstream site. At the downstream and intermediate sites, the buoyancy rate of *S. caseolaris* propagules decreased with an increase in floating time. After 15 days of floating, more than 50% of the propagules sank at both the intermediate site and downstream site, whereas approximately 90% of the propagules at the upstream site remained afloat (Figure 4g).

### 3.4. Root Initiation during Dispersal

Excluding *L. racemosa* and *C. tagal*, the propagules of which did not germinate during the entire floating experiment, the rooting situation of the other mangrove propagules differed significantly among species, floating times, and sites (Table 3). After 20 days of floating, more than 90% of *B. sexangula* and *B. gymnorhiza* propagules rooted at all of the sites, and after 15 days of floating, approximately 85% of *S. alba* propagules rooted at all of the sites. The rooting rates of *R. stylosa*, *A. corniculatum* and *S. caseolaris* were lower than 80% and differed significantly among sites. From upstream to downstream, the final rooting rates of *A. corniculatum* and *S. caseolaris* decreased significantly, while that of *R. stylosa* increased. Of the eight mangrove species, the propagules of *S. alba* germinated fastest. After 1 day of floating, 37.3% of the *S. alba* propagules rooted at the upstream site, followed by *S. caseolaris* and *B. sexangula*, the propagules of which rooted after 3–5 days of floating. *B. gymnorhiza* and *R. stylosa* ranked third. The *A. corniculatum* propagules required 7–10 days to root at the upstream site and intermediate site (Table 3).

**Table 3.** Rooting situation of the propagules of eight mangrove species at three different sites along the Wenchang River. Different letters indicate significant differences at $p < 0.05$ of the same species among sites.

| Species | Root Initiation Time (day) | | | Rooting Rate after 20 Days of Floating (%) | | |
|---|---|---|---|---|---|---|
| | Upstream | Intermediate | Downstream | Upstream | Intermediate | Downstream |
| *Bruguiera sexangula* | 3 | 3 | 5 | 98.3 ± 2.9 a | 95.0 ± 5.0 a | 90.0 ± 5.0 a |
| *Bruguiera gymnorhiza* | 7 | 7 | 7 | 91.7 ± 7.6 a | 95.0 ± 8.6 a | 95.0 ± 5.0 a |
| *Rhizophora stylosa* | 7 | 7 | 7 | 25.0 ± 8.7 a | 21.7 ± 7.6 a | 76.6 ± 16.0 b |
| *Aegiceras corniculatum* | 7 | 10 | - | 70.0 ± 21.8 a | 30.0 ± 18.0 b | 0 |
| *Ceriops tagal* | - | - | - | 0 | 0 | 0 |
| *Lumnitzera racemosa* | - | - | - | 0 | 0 | 0 |
| *Sonneratia caseolaris* * | 3 | 3 | 7 | 60.7 ± 3.1 a | 54.7 ± 6.4 a | 15.3 ± 3.1 b |
| *Sonneratia alba* * | 1 | 1 | 3 | 84.0 ± 2.0 a | 86.0 ± 5.3 a | 84.7 ± 4.2 a |

\* The floating time for *Sonneratia alba* and *Sonneratia caseolaris* was 15 days; -: indicates no root initiation.

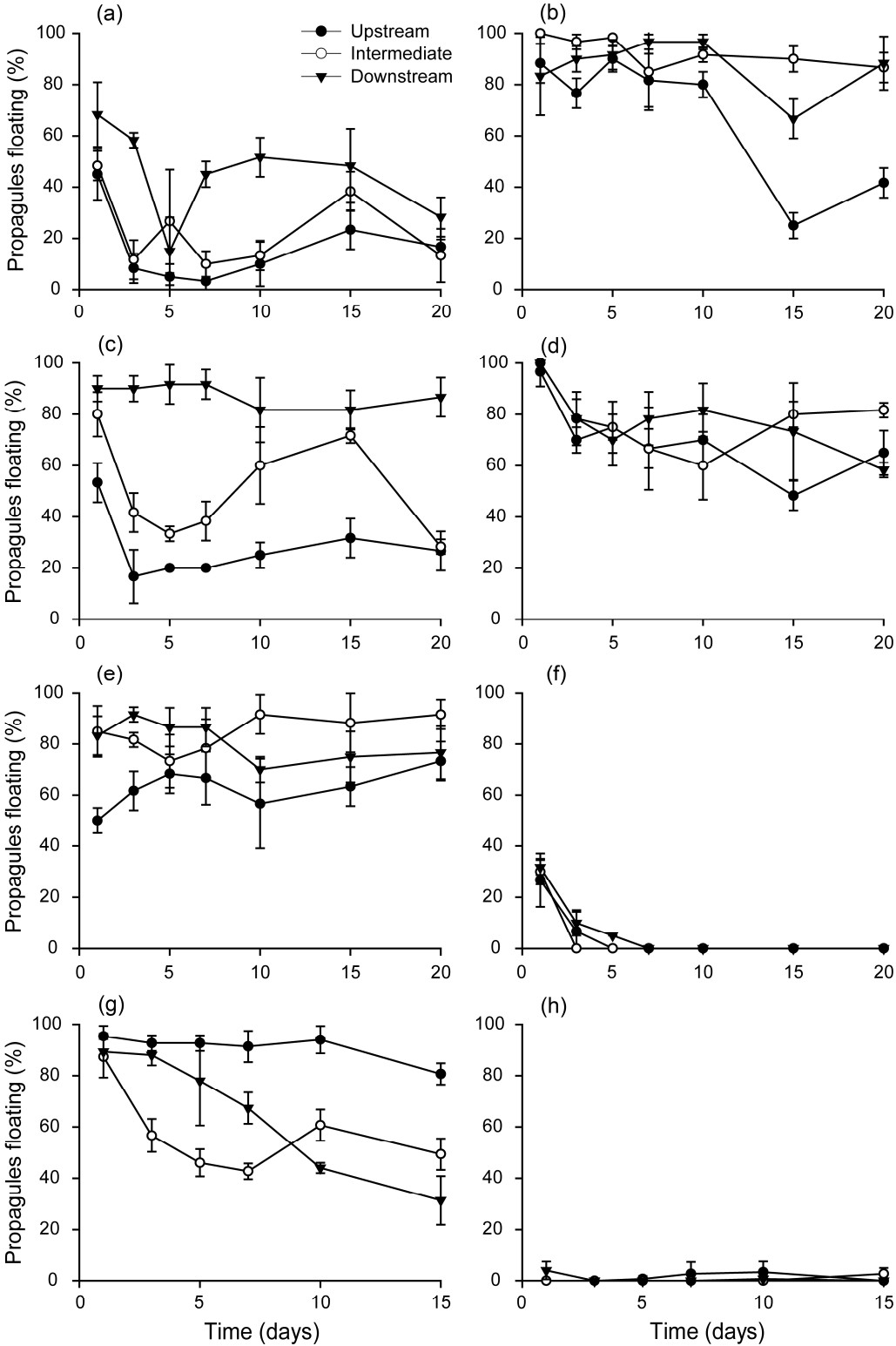

**Figure 4.** Changes in the proportion of the propagules floating of eight mangrove species during floating at three sites with different water salinity regimes. Mean proportion (±SE) shown from 20 propagules placed in each of three replicated grid of woody frame. (**a**) *Bruguiera gymnorhiza*; (**b**) *Ceriops tagal*; (**c**) *Bruguiera sexangula*; (**d**) *Lumnitzera racemosa*; (**e**) *Rhizophora stylosa*; (**f**) *Aegiceras corniculatum*; (**g**) *Sonneratia caseolaris*; (**h**) *Sonneratia alba*.

*3.5. Intertidal and Estuarine Distribution of Mangrove Species*

According to the downstream to upstream distribution pattern, the eight mangrove species could be divided into three categories. Upstream species only occurred in the upstream section, and the typical candidate was *S. caseolaris*. Downstream species only occurred in the downstream section, and the typical candidates included *S. alba*, *R. stylosa*, and *C. tagal*. Whole estuary distribution species occurred from upstream to downstream, and the typical candidates were *A. corniculatum* and *B. sexangula*. They occurred along almost all of the six transects (Figure 5).

According to the surface elevation of the eight mangrove species, *A. corniculatum*, *S. alba*, and *S. caseolaris* could be classified as low intertidal species with mean elevations of lower than 40 cm. *C. tagal* and *L. racemosa* could be classified as high intertidal species, with mean surface elevations higher than 100 cm. *B. gymnorhiza* and *B. sexangula* could be classified as middle intertidal species (Figure 5).

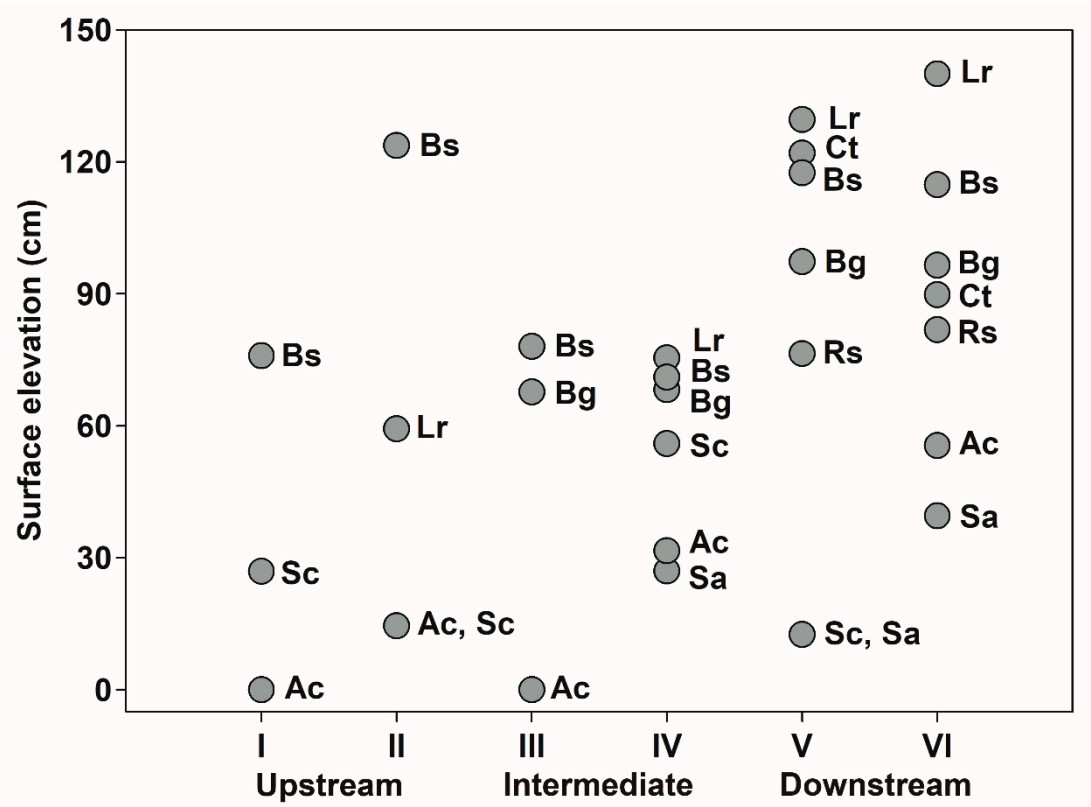

**Figure 5.** Surface elevations of eight mangrove species in different transects along the Wenchang River. Ac: *Aegiceras corniculatum*, Bg: *Bruguiera gymnorhiza*, Bs: *Bruguiera sexangula*, Ct: *Ceriops tagal*, Lr: *Lumnitzera racemosa*, Rs: *Rhizophora stylosa*, Sa: *Sonneratia alba*, Sc: *Sonneratia caseolaris*.

*3.6. Propagule Dispersal Attributes and Tidal Distribution Patterns*

Table 4 shows the linear correlation efficiency between propagule dispersal attributes and tidal distribution patterns. The initial specific gravity was not significantly correlated with the surface elevation. However, the final dynamic specific gravity was negatively correlated with the buoyancy rate and surface elevation ($p < 0.01$). The mangrove species were distributed from the low to high intertidal zone in a manner inversely related to the specific gravity of their propagules.

**Table 4.** Correlations between propagule dispersal attributes and tidal distribution patterns of eight mangrove species along the Wenchang River.

|  | Size (Fresh Weight) | Initial Specific Gravity | Final Dynamic Specific Gravity * | Final Buoyancy Rate * | Final Rooting Rate * |
|---|---|---|---|---|---|
| Initial specific gravity | 0.659 |  |  |  |  |
| Final dynamic specific gravity | −0.294 | −0.022 |  |  |  |
| Final buoyancy rate | 0.129 | −0.099 | −0.833 ** |  |  |
| Final rooting rate | 0.373 | 0.268 | 0.455 | −0.543 |  |
| Surface elevation | 0.470 | 0.407 | −0.849 *** | 0.680 | −0.217 |

* The final dynamic specific gravity, final buoyancy rate and rooting rate of the propagules of eight mangrove species were the data after 15 days or 20 days of floating. **: significant at $p < 0.05$; ***: significant at $p < 0.01$.

## 4. Discussion

### 4.1. Effect of Water Salinity on Propagule Buoyancy and Propagule Changes during Dispersal

Many mangrove propagules have higher buoyancy under high water salinity due to the higher specific gravity of the water [23,46]. Our floating experiments under different water salinity regimes showed that mangrove propagules floated for longer under higher water salinities (Table 2 and Figure 4). The water salinity at the downstream site was higher than at the intermediate and upstream sites. Furthermore, the propagule buoyancy of *B. gymnorhiza* and *B. sexangula* at the downstream site was higher than that at the intermediate site, followed by the upstream site. Van der Stocken et al. [31] reported similar results. Clarke and Myerscough [46] found that *Avicennia* propagules sank faster in brackish water than in saltwater.

A few studies showed that the buoyancy of mangrove propagules decreased with the increase in floating time [29], while the floating orientation could move between horizontal and vertical [23,35,47], which indicates a change in propagule specific gravity during floating. Moreover, some observations even recorded re-buoyancy (regaining buoyancy after an initial sinking) [23,47,48]. Our floating experiment showed similar results. During floating, the densities of the propagules of the eight mangrove species changed dramatically across floating times and sites (Figure 3). In general, the densities of the mangrove propagules increased with floating time despite a small fluctuation. Additionally, our floating experiments showed that the changes in the propagule densities of *B. gymnorhiza*, *R. stylosa*, *S. caseolaris*, and *A. corniculatum* were site specific, which indicated that the changes in their propagule densities were influenced by water salinity.

Since the specific gravity of mangrove propagules changes during dispersal, the dynamic specific gravity is more meaningful in determining propagule dispersal. If the dynamic specific gravity of a propagule is considerably higher or lower than that of seawater, slight changes in the specific gravity of the propagules will have no effect on the buoyancy. These findings have been reported in some mangrove associates [49]. However, when the dynamic specific gravity of the propagules is close to that of seawater, a slight fluctuation in the propagule specific gravity would strongly influence the propagule buoyancy. For example, the dynamic specific gravity of *B. gymnorhiza* and *B. sexangula* initially increased slightly and then decreased, but the proportion of floating propagules under the same condition declined suddenly and then increased (Figures 3 and 4).

Water salinity also has an effect on the root initiation of mangrove propagules. Our floating experiments showed that mangrove propagules delayed root growth under higher salinities (Table 2). Van der Stocken et al. [31] reported similar results. Delayed root initiation under higher salinity was also observed in *Laguncularia racemosa* (L.) Gaertn., *H. littoralis*, and *Acanthus ilicifolius* L. [40].

These results indicate that the dispersal of mangrove propagules is not a passively buoyant process controlled by water currents. During dispersal, mangrove propagules can actively adjust their specific gravity. Water salinity not only has a direct effect on propagule floating but also has an effect on the changes in propagule specific gravity and subsequent floating. Sousa et al. [1] concluded that the interaction between propagule buoyancy and salinity can potentially affect dispersal distance and direction, particularly in estuarine systems where marked gradients in salinity and tidal eddies are

common. Our results are consistent with their conclusion. Additionally, root initiation is generally recognized as a post-dispersal process for terrestrial plants [10,17]. However, our results indicate that root initiation is also a dispersal process in mangrove species the propagules of which are dispersed by water. These support our first hypothesis that the dispersal of mangrove propagules is not a passive dispersal process. Our results differ from those reports where the specific gravity of mangrove propagules during dispersal has been arbitrarily considered as unchangeable [18,30,32].

### 4.2. Propagule Specific Gravity and Root Initiation Determine Establishment

Johansson et al. [50] suggested that propagule specific gravity is an important factor affecting dispersal and establishment in frequently flooded environments. Unfortunately, they did not explain how propagule specific gravity determined dispersal and establishment. Successful establishment at a new location depends on whether the requirements for establishment are met, i.e., the presence of an inundation-free period, the presence of roots that are long enough to resist hydrodynamic forces, and even the presence of longer roots to withstand high-energy events [51].

It is widely believed that tidal currents deliver propagules of all mangrove species to all sections of the intertidal zone [52]. Observations of *R. stylosa* showed that its hypocotyls were found across the entire intertidal gradient [53]. However, the result would be different if more intertidal wetland species, including species with lower specific gravities, were considered. We did not measure the numbers of stranded propagules across the intertidal gradients but found no propagules of *C. tagal* and *L. racemosa* at the low intertidal zone. The propagules with low specific gravity remained buoyant for a relatively long time and tended to become stranded at the upper tidal limits [36]. Most propagules, such as *Xylocarpus*, are often deposited and germinate on the upper beach between spring tides, and a tsunami wave will carry floating propagules a mile or two inland or to further upriver systems [54].

Proper propagule germination and early seedling establishment are vital for seedling survival [55]. These processes require suitable conditions for root penetration into soils [51]. To establish in intertidal zones, mangrove propagules must germinate and root rapidly to avoid being washed away by the next high tide or king tide [54]. In addition to suitable physio-chemical factors for root growth, propagules must be in contact with the soil for a certain amount of time for establishment to occur. During this process, hydrodynamic force is a major threat to seedling establishment [31,51]. With the increase in surface elevation from the low intertidal zone via the middle intertidal zone to the high intertidal zone, water inundation frequency, water velocity, and wave energy decrease, which results in a decrease in hydrodynamic force. The presence of an inundation-free period and the presence of roots to resist water turbulence and tidal action have been stated as two prerequisites for the successful establishment of a mangrove propagule [31,51]. These two prerequisites are especially needed for species with a higher intertidal distribution. These species include all associates and mangrove species the dynamic specific gravity of which is lower than 1.00 g/cm$^3$ (Table 1). Consider *L. racemosa* as an example: its initial propagule specific gravity was 0.868 g/cm$^3$ (Table 1), and its mean surface elevation was approximately 115 cm, which indicates a higher intertidal distribution and flooding only during spring tides. Most propagules may be stranded in the vicinity of their parent trees [29]. These stranded propagules will only be floated during incoming spring tides and require 60 days for seed germination [56]. The period for *Lumnitzera littorea* (Jack) Voight. propagule germination is 44 days [57]. The above results indicate that a low germination rate is a general characteristic of backshore species [23]. In this area, the inundation-free period is long enough for these propagules to germinate without being disturbed by tides. Before the next spring tide, their roots are long enough to resist hydrodynamic forces [51]. Additionally, even if their propagules were distributed in the mid-intertidal zone or the low intertidal zone by chance, the hydrodynamic force resulting from the large differences between propagule specific gravity and water specific gravity, as well as water inundation once or twice daily, makes it impossible for the low-specific gravity propagules to use the short inundation-free period for establishment. These propagules will be carried away by incoming tides, and even though root initiation occurs in some propagules, delicate roots may be damaged by

water turbulence [55]. Additionally, a high water inundation frequency in such areas has a negative physiological effect on root initiation [58].

After reaching a new location by dispersal, the early growth stage of mangrove propagules influences the spatial formation of mangrove zonation. Before establishment, processes of predation may influence the initial patterns of distribution [27,52] because predator numbers are generally lowest in the lowest intertidal zone and increase to maximum amounts in the high intertidal zone [59,60]. After successful establishment, a number of physico-chemical factors, such as salinity, waterlogging, the physical and chemical properties of the soil, and light, may affect seedling growth and survival [61]. Regarding pioneer species such as *A. marina* and *S. alba*, it was hypothesized that shade intolerance, lack of seed dormancy, high tolerance to salinity, and regular inundation, as well as resistance to wave action, were the traits that facilitated establishment at lower intertidal zones [62]. However, this explanation did not account for how establishment occurs in lower intertidal zones or the ability of the propagule to resist wave action. It was explained that establishment occurs during neap tides when water inundation does not occur [31,51]. Balke et al. [51] emphasized that fast root growth during establishment is particularly important for anchoring the seedling sufficiently so as to withstand inundation and wave action, especially in the low intertidal zone. Our results are partly consistent with this. For example, the final dynamic specific gravity of the *S. alba* propagule increased to 1.12 g/cm$^3$ after a few days of floating, and most of the propagules rooted and sank at the downstream site (Figure 3 and Table 2). A simulation experiment demonstrated that it took two days for the root primordia to be visible after stranding, and on the third day, the maximum root length was 1 cm [51]. Except under some abnormal conditions, such as strong northerly winds typical of Louisiana winters [48], it is impossible for the propagule to use this inundation-free period to complete root development in low intertidal zones that are flooded once or twice daily. In these locations, the large specific gravity differences between the propagules and water allows these propagules to resist wave forces. Since the propagule specific gravity of pioneer species is higher than that of the water, the stranded propagules may come into contact with the soil and thus possibly resist the incoming tides. Additionally, both theoretical analysis and field experiments showed that the combined effects of drag caused by mangrove roots and bottom friction produced a significant amount of attenuation over a relatively short distance. From the bottom to the water surface, the wave energy attenuated sharply. Even during high tides, the wave energy at the water bottom was close to zero [63]. High salt tolerance and high waterlogging tolerance ensure growth at this type of sites [21]. In conclusion, for propagules with low specific gravity, a period that is free of water inundation is critical for establishment, and for propagules with a specific gravity higher than water, the specific gravity difference between the propagules and water is critical for their establishment.

From downstream to upstream, water salinity decreases. Species with higher propagule specific gravity, such as *S. alba* will sink rapidly. There is no chance for drift to upstream regions. Although the propagules of *S. caseolaris* have the chance to drift to downstream sections during ebb tide and then sink, root germination and establishment will be inhibited by the higher water salinity due to the low salt tolerance of this species [25]. This explains the absence of *S. caseolaris* in the downstream regions and the absence of *S. alba* in the upstream regions.

Smith [52] inferred that the critical process influencing mangrove zonation is not propagule dispersal, but rather post-dispersal establishment, survival, competition, and growth. Numerous studies have shown that propagule dispersal is much more important than post-dispersal establishment [4,12,16,17]. Our results show that mangrove zonation is determined by both propagule dispersal and early establishment; however, propagule dispersal is more important. Additionally, the tidal sorting hypothesis not only acts on propagule dispersal, but also acts on the early establishment of mangrove propagules. After establishment, survival and growth are strongly influenced by physio-chemical stresses, animal predation, and competition [58,61,64].

Mangrove forests are highly susceptible to global climate changes (e.g., precipitation change, increasing storm intensity, accelerating sea level rise, land subsidence and changing sediment

supply) [65–67], which will influence the water salinity and surface elevation of mangrove forests. Our findings show that the mangrove zonation is determined by propagule dispersal, which is controlled by water salinity and surface elevation. Therefore, we can infer that global change could alter the way propagules disperse by changing water salinity and surface elevation, and then influence the zonation of mangrove forests.

## 5. Conclusions

We conclude that the dispersal of mangrove propagules is not a passive but an active process. Water salinity is a vital factor that affects the buoyancy characteristics of mangrove propagules, including the flotation period, viability, and root initiation. Our results support Rabinowitz's conclusion that mangrove zonation on both the intertidal and estuarine scale can be explained by tidal sorting hypothesis. However, we disagree with Rabinowitz's explanation that mangrove zonation is controlled by "tidal sorting of the propagules according to size" [44]. Mangrove zonation is controlled by the tidal sorting of the propagules according to buoyancy and by the differential ability of propagules to establish in the intertidal zone. These dispersal behaviors agree well with the distribution patterns of the species across the estuary and the intertidal zone in the study site. Mangrove propagule establishment in the intertidal zone is controlled by specific gravity, root initiation time and water inundation frequency. Our results support the "supply-side theory" that mangrove zonation is more strongly explained by the supply of new propagules arriving and recruiting to a site than by post-recruitment biotic interactions. Each mangrove species finds its place along the environmental gradients resulting from the tidal level and the water salinity, which determine propagule anchoring.

The results of our study add new understanding of observed mangrove species zonation and should inform conservation managers when restoring mangroves or evaluating the potential impacts of anthropogenic (e.g., dam construction and seawall construction) and natural disturbances that might alter the hydrology, including the water salinity regime.

**Author Contributions:** M.W. and W.W. conceived the ideas, designed the experiments, and wrote the manuscript. X.L. collected and analyzed the data. All authors read and approved the final manuscript.

**Funding:** This work was jointly supported by grants from the Programs of Science and Technology on Basic Resources Survey for the Ministry of Science and Technology of China (2017FY100701), and the National Natural Science Foundation of China (31670490).

**Acknowledgments:** We sincerely thank Norman C. Duke for constructive suggestion during manuscript preparing and we would like to thank LetPub (www.letpub.com) for providing linguistic assistance during the preparation of this manuscript.

**Conflicts of Interest:** The authors declare no conflicts of interest.

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
