# Peer review of "Propagule Dispersal Determines Mangrove Zonation at Intertidal and Estuarine Scales"

_forests, doi:10.3390/f10030245_

Reviewer 1 Report

Overall manuscript is quite interesting, timely and add to new knowledge and understanding of propagule dispersal and its role in mangrove zonation.

Below are my comments for improving the manuscript

Throw-out the manuscript author need to differentiate propagule from seed.  Propagule should be used where author is referring to mangroves as propagules production is a part of viviparous germination which found in mangroves when attached to mother plant. On the other hand seed germinate after detached from the mother tree. So, terminology should be clear. When author referring to tropical plant or forest that includes all kind of plant and forest in that mangroves are a kind.    

Avoid repetition, lot of places author have mentioned same thing like total 8 species were chosen, 20 propagules were selected etc.  

Better to use upstream, intermediate and downstream rather than site 1, 2 and 3, respectively. As author have only three sites and it will be easy for readers to understand via up, intermediate and downstream rather than site names. Table 3 author have used upstream, intermediate and downstream term and readers do not need to see map again and again.     

Density term have been used in manuscript. Density (no. per unit area) or (mass/volume). Authors need to use different word of each term better to use specific gravity rather than density. Line 346 – 347 confusing which density authors are referring here.   

 Abstract

Line 10 = Change propagule to seed because author is taking tropical forest in account not mangroves

Line 28-30 = Research can also be helpful in restoring mangroves naturally, authors need to add this part in conclusion section

Introduction

Line 44 – 54 = Authors referring to propagule buoyancy and it’s a part of propagule characteristics therefore should come after mangrove zonation paragraph (Shift before line 71)

Line 90-92 = I think mangrove zonation on intertidal zone is quite explained in literature such as from seaward side to landward side and salinity is an important factor determining this zonation. Which depend on species growing in particular zone depend on salt tolerant capacity of that species. Rewrite this sentence.

Line 135 – authors need to write full name both genus and species as some Heritiera are true mangroves.

Line 136 – What is the distance between the plots? How long was each transect? Authors need to provide this information.

Line 137-138 – Any particular definition for mature trees and shrubs (references??)

Line 143 – Reference for formula used to estimate relative modal elevation? Authors need to provide more explanation for this formula and how tree density account in measurement of elevation?

Line 152 – Authors have reported only 7 species in paragraph (Line 112 – 123) but here 8 dominant species (Clarify this part) – Line 160 Ceriops species is mentioned (It is upstream intermediate or downstream)

Line 294 – it’s a repletion of sentence that eight mangrove species were used. Authors have already mentioned it earlier. Delete sentence or rewrite

Line 305 – 306 – Rewrite this sentence. Difficult to understand

Line 416 – 420- very long sentence, need to rewrite

Author Response

Dear Sir,

Thank you very much for your consideration of our manuscript. The comments are very constructive. We have revised the whole manuscript carefully and tried to avoid any grammar or syntax error. The revised manuscript has been proof-read by professional language office Letpub (www.letpub.com). The followings are our detailed responses to each comment.

All detailed corrections are listed below point by point:

1.     Throw-out the manuscript author need to differentiate propagule from seed. Propagule should be used where author is referring to mangroves as propagules production is a part of viviparous germination which found in mangroves when attached to mother plant. On the other hand seed germinate after detached from the mother tree. So, terminology should be clear. When author referring to tropical plant or forest that includes all kind of plant and forest in that mangroves are a kind.

Partially accepted and revised accordingly. Mangrove propagules include seed, fruit and seedling (Tomlinson, 2017). Not all mangrove species are viviparous. In our study, the propagules of Sonneratia alba and Lumnitzera racemosa are seed and fruit, respectively. To avoid misunderstanding, we reorganized Chapter 2.3 accordingly. See Line 188-203.

2.     Avoid repetition, lot of places author have mentioned same thing like total 8 species were chosen, 20 propagules were selected etc.

Accepted and revised accordingly.

3.     Better to use upstream, intermediate and downstream rather than site 1, 2 and 3, respectively. As author have only three sites and it will be easy for readers to understand via up, intermediate and downstream rather than site names. Table 3 author have used upstream, intermediate and downstream term and readers do not need to see map again and again.

Accepted and revised accordingly. See Fig. 1, Fig.2, Fig. 3 and Fig. 4

4.     Density term have been used in manuscript. Density (no. per unit area) or (mass/volume). Authors need to use different word of each term better to use specific gravity rather than density. Line 346 – 347 confusing which density authors are referring here.

Accepted and revised accordingly. The term “density (mass/volume)” was changed to “specific gravity”.

5.     Line 10 = Change propagule to seed because author is taking tropical forest in account not mangroves

Do not accept. The definition of plant propagule: a plant part that becomes detached from the parent tree and grows into new plant. Seed is only a type of plant propagule. Plant propagules can be classified into spore, seed, fruit, seedling and cutting.

6.     Line 28-30 = Research can also be helpful in restoring mangroves naturally, authors need to add this part in conclusion section

Accepted and revised accordingly. See Line 30-31.

7.     Line 44 – 54 = Authors referring to propagule buoyancy and it’s a part of propagule characteristics therefore should come after mangrove zonation paragraph (Shift before line 71)

Accepted and revised accordingly. See Line 48-85.

8.     Line 90-92 = I think mangrove zonation on intertidal zone is quite explained in literature such as from seaward side to landward side and salinity is an important factor determining this zonation. Which depend on species growing in particular zone depend on salt tolerant capacity of that species. Rewrite this sentence.

Accepted and revised accordingly. See Line 107-113.

9.     Line 135 – authors need to write full name both genus and species as some Heritiera are true mangroves.

Accepted and revised accordingly. See Line 163-164.

10.  Line 136 – What is the distance between the plots? How long was each transect? Authors need to provide this information.

Accepted and revised accordingly. The distance between the adjacent quadrat was 20m. The length of the transects ranged between 70m and 300 m. See Line 158-160.

11.  Line 137-138 – Any particular definition for mature trees and shrubs (references??)

Accepted and revised accordingly. See Line 168.

Fang, J.Y.; Wang, X.P.; Shen, Z.H.; Tang, Z.Y.; He, J.S.; Yu, D.; Jiang, Y.; Wang, Z.H.; Zheng, C.Y.; Zhu, J.L.; Guo, Z.D.; Methods and protocols for plant community inventory. Biodivers. Sci. 2009, 17, 533–548.

12.  Line 143 – Reference for formula used to estimate relative modal elevation? Authors need to provide more explanation for this formula and how tree density account in measurement of elevation?

Accepted and revised accordingly. See Line 186. We added two references.

Oh, R.R.Y.; Friess, D.A.; Brown, B.M. The role of surface elevation in the rehabilitation of abandoned aquaculture ponds to mangrove forests, Sulawesi, Indonesia. Ecol. Eng. 2017, 100, 325–334.

Leong, R.C.; Friess, D.A.; Crase, B.; Lee, W.K.; Webb, E.L. High-resolution pattern of mangrove species distribution is controlled by surface elevation. Estuar. Coast. Shelf Sci. 2018, 202, 185–192.

13.  Line 152 – Authors have reported only 7 species in paragraph (Line 112 – 123) but here 8 dominant species (Clarify this part) – Line 160 Ceriops species is mentioned (It is upstream intermediate or downstream)

We checked the species mentioned in the paragraph (Line 113-123). There had 8 species. Ceriops tagal is a downstream species.

14.  Line 294 – it’s a repletion of sentence that eight mangrove species were used. Authors have already mentioned it earlier. Delete sentence or rewrite

Accepted and revised accordingly. See Line 361-362.

15.  Line 305 – 306 – Rewrite this sentence. Difficult to understand

Accepted and revised accordingly. See Line 371-376.

16.  Line 416 – 420- very long sentence, need to rewrite.

Accepted and revised accordingly. See Line 499-500.

Reviewer 2 Report

The present is a simple, but very interesting and well-planned study about the ecology of propagule dispersal in order to explain zonation patterns of adult trees in tropical mangles. It represents a good contribution to the understanding of the ecology of mangroves.

The ecological framework and antecedents of the study problem are very well presented, although I suggest some minor changes (see below). The experimental design is robust, although the replication is a bit small. The statistical is simple but adequate. The results are in general clearly presented (see minor recommendation below). The results are well analyzed and interpreted, and deeply discussed in the frame of the main antecedents. The main conclusions are supported by the results of the study (see minor recommendation below).

Recommendations

1)      The introduction is correct, but I recommend start by describing the ecological problem under study, i.e., mangrove zonation patterns (Lines 55 to 92), and then introduce the antecedents supporting your hypothesis (Lines 1-54).

2)      Methodology: The section “Measurement of tidal elevation” is not clear enough to mi. How are located the transects and sampling stations, with respect to sea level? Please explain this.

3)      Results: The section “Intertidal and estuarine distribution of mangrove species” should be supported with data (table or fig) about mean tidal elevation of species. Please present supporting data for this.

4)      Conclusion: “After establishment, survival and growth are strongly influenced by physiochemical stresses, animal predation and competition.”. In this study biological interactions (predation, competition) were not addressed. It would be better to focus on those conclusions derived specifically on your results and discuss previous results (other papers) in the Discussion.

Author Response

Dear Sir, Thank you very much for your consideration of our manuscript. The comments are very constructive and we have revised the whole manuscript carefully. The followings are our detailed responses to each comment. 1. The introduction is correct, but I recommend start by describing the ecological problem under study, i.e., mangrove zonation patterns (Lines 55 to 92), and then introduce the antecedents supporting your hypothesis (Lines 1-54). Partially accepted and revised accordingly. See Line 1-113. 2. Methodology: The section “Measurement of tidal elevation” is not clear enough to mi. How are located the transects and sampling stations, with respect to sea level? Please explain this. Accepted and revised accordingly. See Line 174-186. We added two references. Oh, R.R.Y.; Friess, D.A.; Brown, B.M. The role of surface elevation in the rehabilitation of abandoned aquaculture ponds to mangrove forests, Sulawesi, Indonesia. Ecol. Eng. 2017, 100, 325–334. Leong, R.C.; Friess, D.A.; Crase, B.; Lee, W.K.; Webb, E.L. High-resolution pattern of mangrove species distribution is controlled by surface elevation. Estuar. Coast. Shelf Sci. 2018, 202, 185–192. 3. Results: The section “Intertidal and estuarine distribution of mangrove species” should be supported with data (table or fig) about mean tidal elevation of species. Please present supporting data for this. Accepted and revised accordingly. We added the surface elevation data of the 8 species (Figure 5). See Line 355-358. 4. Conclusion: “After establishment, survival and growth are strongly influenced by physiochemical stresses, animal predation and competition.”. In this study biological interactions (predation, competition) were not addressed. It would be better to focus on those conclusions derived specifically on your results and discuss previous results (other papers) in the Discussion. Accepted and revised accordingly. We deleted the last sentence. See Line 547-548.
